# Ecological networks are more sensitive to plant than to animal extinction under climate change

Matthias Schleuning[1], Jochen Fründ[2,3], Oliver Schweiger[4], Erik Welk[5,6], Jörg Albrecht[7,8], Matthias Albrecht[9], Marion Beil[10], Gita Benadi[3], Nico Blüthgen[11], Helge Bruelheide[5,6], Katrin Böhning-Gaese[1,12], D. Matthias Dehling[1,13], Carsten F. Dormann[3], Nina Exeler[14], Nina Farwig[7], Alexander Harpke[4], Thomas Hickler[1,15], Anselm Kratochwil[14], Michael Kuhlmann[16,17], Ingolf Kühn[4,5,6], Denis Michez[18], Sonja Mudri-Stojnić[19], Michaela Plein[20], Pierre Rasmont[18], Angelika Schwabe[10], Josef Settele[4,6], Ante Vujić[19], Christiane N. Weiner[11], Martin Wiemers[4] & Christian Hof[1]

Impacts of climate change on individual species are increasingly well documented, but we lack understanding of how these effects propagate through ecological communities. Here we combine species distribution models with ecological network analyses to test potential impacts of climate change on >700 plant and animal species in pollination and seed-dispersal networks from central Europe. We discover that animal species that interact with a low diversity of plant species have narrow climatic niches and are most vulnerable to climate change. In contrast, biotic specialization of plants is not related to climatic niche breadth and vulnerability. A simulation model incorporating different scenarios of species coextinction and capacities for partner switches shows that projected plant extinctions under climate change are more likely to trigger animal coextinctions than vice versa. This result demonstrates that impacts of climate change on biodiversity can be amplified via extinction cascades from plants to animals in ecological networks.

[1] Senckenberg Biodiversity and Climate Research Centre (BiK-F), Senckenberganlage 25, 60325 Frankfurt am Main, Germany. [2] Department of Integrative Biology, University of Guelph, Guelph, Ontario, Canada N1G2W1. [3] Biometry and Environmental Systems Analysis, University of Freiburg, Tennenbacherstr. 4, 79106 Freiburg, Germany. [4] Helmholtz Centre for Environmental Research–UFZ, Department of Community Ecology, Theodor-Lieser-Str. 4, 06120 Halle, Germany. [5] Institute for Biology, Martin Luther University Halle-Wittenberg, Am Kirchtor 1, 06108 Halle (Saale), Germany. [6] German Centre for Integrative Biodiversity Research (iDiv) Halle-Jena-Leipzig, Deutscher Platz 5e, 04103 Leipzig, Germany. [7] Conservation Ecology, Faculty of Biology, Philipps-Universität Marburg, Karl-von-Frisch-Str. 8, 35032 Marburg, Germany. [8] Institute of Nature Conservation, Polish Academy of Sciences, Mickiewicza 33, 31-120 Kraków, Poland. [9] Institute for Sustainability Sciences, Agroscope, Reckenholzstr. 191, CH-8046 Zürich, Switzerland. [10] Vegetation and Restoration Ecology, Department of Biology, Technische Universität Darmstadt, Schnittspahnstr. 10, 64287 Darmstadt, Germany. [11] Ecological Networks, Department of Biology, Technische Universität Darmstadt, Schnittspahnstr. 3, 64287 Darmstadt, Germany. [12] Department of Biological Sciences, Johann Wolfgang Goethe University of Frankfurt, Max-von-Laue-Str. 9, 60438 Frankfurt am Main, Germany. [13] School of Biological Sciences, University of Canterbury, Private Bag 4800, Christchurch 8140, New Zealand. [14] Ecology Section, Department of Biology and Chemistry, University of Osnabrück, Barbarastr. 13, 49076 Osnabrück, Germany. [15] Department of Physical Geography, Geosciences, Johann Wolfgang Goethe University of Frankfurt, Altenhöferallee 1, 60438 Frankfurt am Main, Germany. [16] Zoological Museum, University of Kiel, Hegewischstr. 3, 24105 Kiel, Germany. [17] Department of Life Sciences, The Natural History Museum, Cromwell Road, London SW7 5BD, UK. [18] Laboratory of Zoology, Biosciences Institute, University of Mons, Place du Parc 20, B-7000 Mons, Belgium. [19] Department of Biology and Ecology, Faculty of Sciences, University of Novi Sad, Trg Dositeja Obradovića 2, 21000 Novi Sad, Serbia. [20] Geography Planning and Environmental Management, The University of Queensland, St Lucia, Queensland 4067, Australia. Correspondence and requests for materials should be addressed to M.S. (email: matthias.schleuning@senckenberg.de).

Climate change forces species either to move or to adapt to changing conditions[1,2]. Although models predicting the responses of individual species to climate change are widely utilized[2], it is not yet clear to what extent a changing climate will affect biotic interactions between species[3,4]. Ecological theory predicts that abundant generalist species tend to have large ranges[5] and, consequently, occupy wide climatic niches[6], whereas species specialized on specific interaction partners have small ranges, occupy narrow climatic niches and may therefore be particularly vulnerable to climate change[7].

In ecological communities, species are embedded in networks of interacting species, for instance, in mutualistic networks between plant species and animal pollinators or seed dispersers[8]. Species in these networks vary in the number of interaction partners, for example, because of differences in species traits[9], and thus differ in their degree of biotic specialization[8]. So far, it has not been tested how biotic specialization in ecological networks relates to a species' climatic niche breadth and its vulnerability to climate change. However, a quantitative understanding of this relationship is required to predict the likelihood of species extinctions and coextinctions from ecological communities under climate change[10].

Here we test the two hypotheses that plants and animals with (1) narrow climatic niches and (2) a projected loss in climatic suitability are biotic specialists that interact with a low diversity of partners. We additionally simulate (3) how the relationship between biotic specialization and vulnerability to climate change affects the risk of species coextinctions of plants and animals under future climatic conditions. We analysed data on climatic niche breadth for 295 species of plants and their insect pollinators (196 bee, 70 butterfly and 97 hoverfly species) and seed dispersers (51 bird species) from central Europe. For each species, we quantified the change in climatic suitability across a species' current European range under projected climate change according to two circulation models and two representative concentration pathways (RCPs 6.0 and 8.5). We linked projected changes in climatic suitability to data on biotic specialization derived from 8 quantitative pollination and 5 quantitative seed-dispersal networks recorded in 13 regions across central Europe. Networks describe interaction frequencies between plant and animal species, that is, the number of visits of an animal to a plant species, and yield empirical estimates of biotic specialization for each species in each network.

We find that animal species with narrow climatic niches and a projected loss in climatic suitability interact with a low diversity of plant partners, whereas we do not find analogous relationships for plants. This important difference between plant and animal species affects the likelihood of species coextinctions under climate change. We simulate different scenarios of species coextinction and capacities for partner switches and show with these simulations that mutualistic networks are more sensitive to projected plant than to animal extinctions under climate change. We conclude that a high potential for adaptive partner switches is required to stabilize mutualistic networks against extinction cascades from plants to animals under climate change.

## Results

**Climatic niche breadth and biotic specialization.** In line with the first hypothesis, we found that animals' climatic niche breadth was positively associated with the effective number of plant partners in the regional pollination and seed-dispersal networks (Fig. 1a,b). In contrast, climatic niche breadth of plants was not related to the effective number of animal partners (Fig. 1a,b). For both plants and animals, we found no relationship between climatic niche breadth and complementary specialization $d'$

(a measure of the uniqueness of interaction partners relative to other species; Supplementary Table 1). These trends were qualitatively similar across the individual networks (Supplementary Table 2).

**Vulnerability to climate change and biotic specialization.** Consistent with the second hypothesis, we found that animals projected to lose climatic suitability across their current European range had a low diversity of plant partners in the regional networks (Fig. 1c,d). There was no analogous relationship for plants and their effective number of animal partners (Fig. 1c,d). Changes in climatic suitability were unrelated to complementary specialization $d'$ for both plants and animals (Supplementary Table 1). Trends were again similar across the individual networks (Supplementary Table 2).

**Species coextinctions under climate change.** We simulated secondary extinctions of animal and plant species from mutualistic networks as a consequence of sequential species loss from the other trophic level. Extending upon previous simulations of species coextinctions[11–13], we informed our simulation model with projected changes in climatic suitability for plant and animal species and removed species sequentially from the highest to the lowest decrease in climatic suitability (Fig. 2). We modelled species coextinction under different scenarios of species' sensitivity to partner loss assuming that a 25, 50 or 75% decrease in total interaction frequency would trigger the secondary extinction of a species from the regional network. These thresholds are probably more realistic than the assumption that all interaction partners must be lost to trigger secondary extinction[11–13], given the frequency of coextinctions reported in empirical and modelling studies[14,15]. In the simulation, we further accounted for the potential flexibility of species in the choice of their interaction partners by reallocating a varying proportion of lost interaction events to persisting species. We account for a potential rewiring of interactions to new partners[13,16] by comparing a scenario of constrained rewiring to persisting partners with a scenario of unconstrained rewiring to all persisting species. We did not consider, however, that new species may enter the interaction networks and, thus, may overestimate extinction risks under climate change[17]. Furthermore, we assumed that interaction frequencies indeed reflect the reciprocal functional dependences of animals on plants and vice versa[18]. For each network and simulation scenario, we quantified the relationship between primary and secondary species extinctions, yielding a measure of network sensitivity to plant and animal extinction, respectively (Fig. 2). We compared network sensitivity to species coextinction between extinction sequences due to climate change and due to random extinction, thereby accounting for effects that are independent of the extinction sequence, such as inherent differences in species numbers and mean specialization between plants and animals.

Across networks, we found that secondary animal extinctions were more likely to occur than secondary plant extinctions (Fig. 3 and Supplementary Fig. 1). In almost all scenarios, this difference was larger under climate change than under random extinction (Fig. 3) independent of the chosen RCP scenario (Supplementary Fig. 2). Differences between climate change and random extinction were most pronounced if we assumed a high species' sensitivity to coextinction and a low capacity for rewiring lost interactions to other partners (see, for example, Fig. 3a). Scenarios in which many interactions needed to be lost to trigger secondary extinction differed less between climate change and random extinction, especially if species were able to reallocate many of their lost interactions to persisting species in the network (see, for

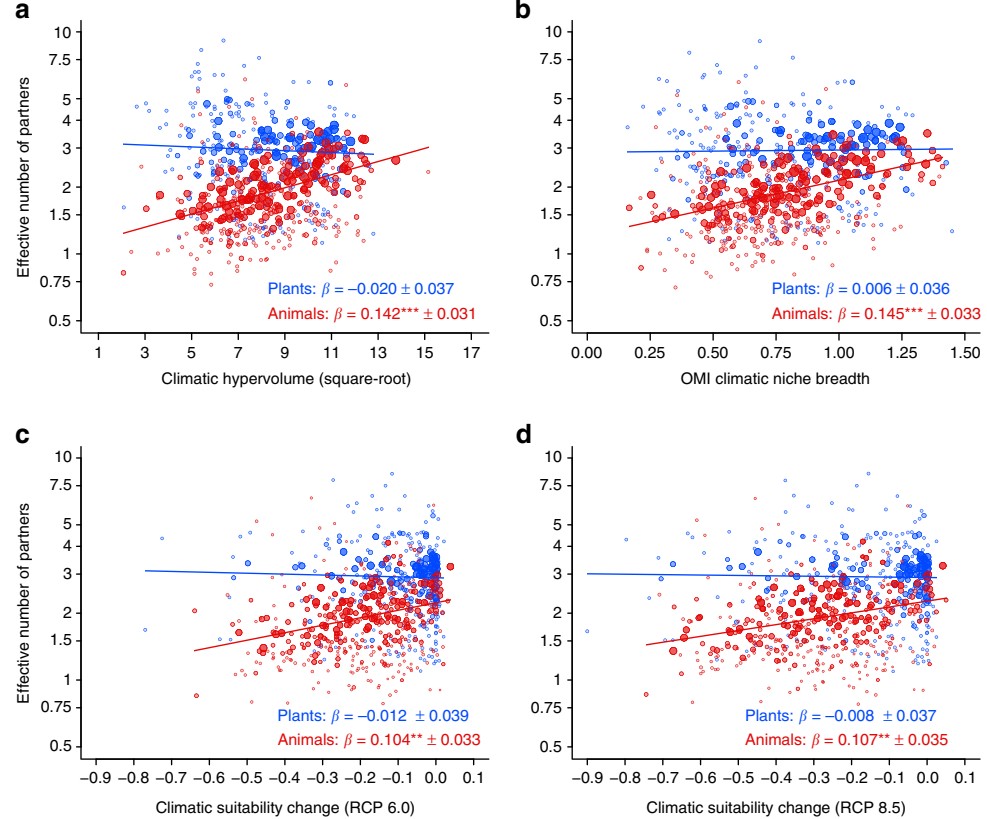

**Figure 1 | Biotic specialization in relation to climatic niche breadth and vulnerability to climate change.** Associations of (**a,b**) realized climatic niche breadth (climatic hypervolume[60], OMI climatic niche breadth[61]) and (**c,d**) projected climatic suitability change (RCP 6.0, RCP 8.5 scenarios[65]; year 2070) with the effective number of partners ($e^H$) of plant ($n = 295$) and animal ($n = 414$) species in 13 mutualistic interaction networks from central Europe. Specialization is the effective number of interaction partners[66] of plant (blue) and animal (red) species in each network (shown on a log-scale). Trend lines indicate the estimated slope ($\beta$) in a mixed-effects model accounting for effects of network identity and animal and plant taxonomy on model intercepts. Shown are species' mean partial residuals plus intercept from these models; symbol size is proportional to the weight of each species in the analysis, corresponding to its number of occurrences across networks and, in the case of climatic suitability change, the accuracy of the species distribution model ($TSS_{max}$ value[64]); given are slope estimates ±1 s.e. for plants and animals, $P$ values were derived by Kenward–Roger approximation: $**P < 0.01$ and $***P < 0.001$ (for full statistics see Supplementary Table 1).

example, Fig. 3f). Hence, animal coextinctions in response to plant extinction were most frequent if animals were limited in their flexibility to respond to future changes in partner availability.

## Discussion

For animals, but not for plants, our results support the first hypothesis that species with narrow climatic niches are biotic specialists. Different, not mutually exclusive, explanations are consistent with this finding. First, animal species with wide distribution ranges and climatic niches are usually locally abundant[19] and are therefore likely to locally interact with more plant partners than rare species. In contrast, the relationship between range size and local abundance is usually more variable for plants[20]. Second, species traits that favour biotic generalization, for example, large body size[21], may also favour the widespread distribution of animal species across a wide climatic range[22]. Thus, climatic niche breadth and biotic specialization may be indirectly linked via species traits. Third, realized climatic niche breadth and biotic specialization will be directly linked if the distribution of specialized animal species is constrained by that of their resource plants, which has been demonstrated for antagonistic plant–animal interactions of butterflies and other phytophagous insects[7,23], but not yet for animal species linked to plants by mutualistic interactions. In contrast to animals, plants

may depend less on their animal partners because pollination and seed dispersal by animals are characterized by a high degree of animal redundancy[24] and because many plants have evolved alternative regeneration loops, such as clonal propagation, autonomous self-pollination or the maintenance of persistent seed banks[25].

In line with our second hypothesis, we found that specialized animals may indeed be more vulnerable to climate change than generalists. Thus, climate change is likely to trigger a decline or even the local extinction of animal species that are constrained by the occurrence of specific plant partners[7,23]. However, as the most connected animals seem to be relatively tolerant to projected changes in climatic conditions, climate change may only have weak indirect impacts on ecological networks via top-down effects from animals to plants. In contrast to animals, we did not detect a relationship between biotic specialization and vulnerability to climate change for plants in central Europe. This result suggests that highly connected plants are similarly threatened as weakly connected plants. The decline or loss of highly connected plants that interact with many animal partners could have important bottom-up impacts on animal species and ecological networks[26,27].

Simulations of species coextinctions indeed demonstrate that mutualistic networks are more sensitive to plant extinction than to animal extinction under climate change in central Europe. This effect could be related to two different mechanisms. First, most

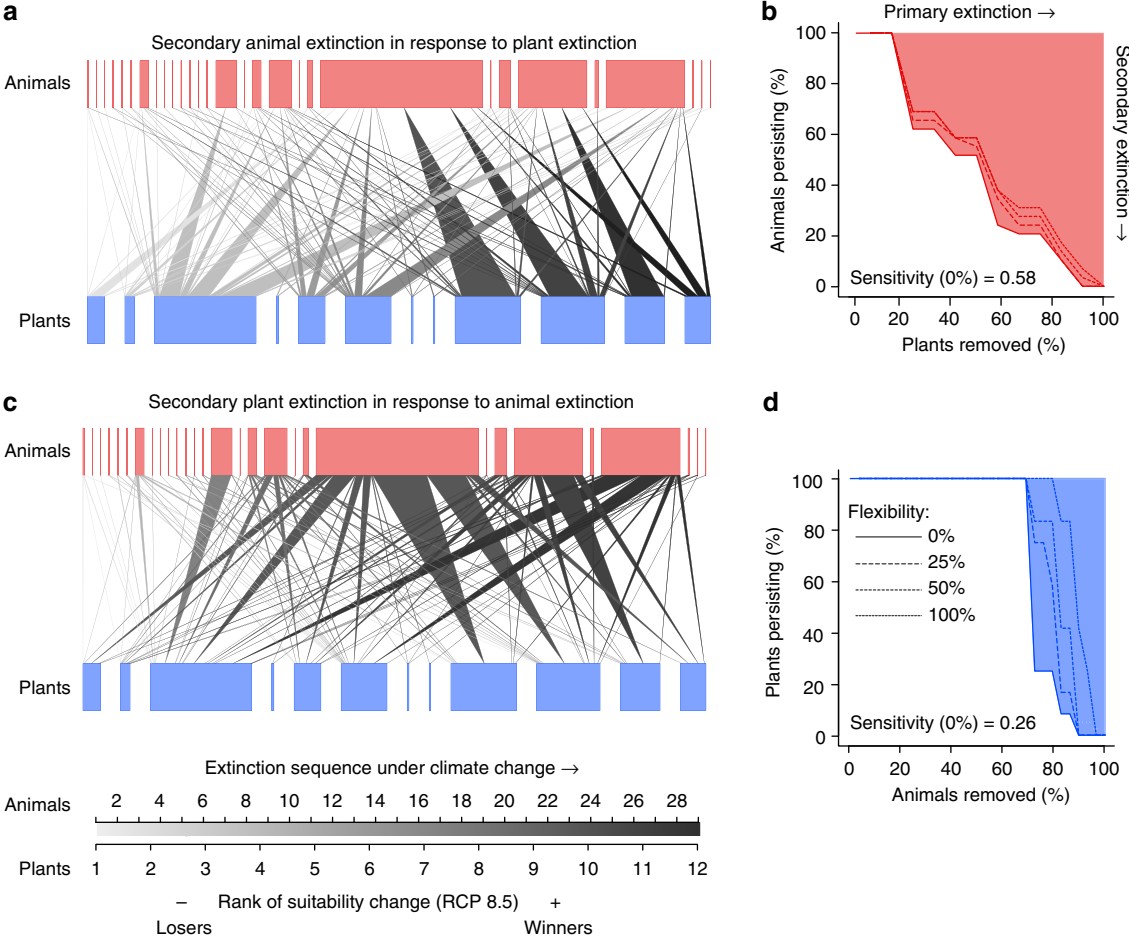

**Figure 2 | Secondary animal and plant extinction under climate change.** Shown are (**a,b**) secondary animal extinction in response to plant extinction and (**c,d**) secondary plant extinction in response to animal extinction for a seed-dispersal network from Białowieża forest (network ID = S1; 12 plant and 29 bird species). (**a,c**) Species (rectangles in red (animals) and blue (plants), connected by weighted interaction links; box and line width correspond to interaction frequencies) are removed sequentially according to projected suitability changes in climatic conditions. Low ranks (light shade) correspond to a high vulnerability to climate change, high ranks (dark shade) correspond to a low vulnerability; thus, light links are prone to extinction, whereas dark links are the persisting backbone of interactions under climate change. The corresponding secondary extinction plots (**b**) for animals (red) and (**d**) plants (blue) show network sensitivity to species extinction (filled area above the extinction curve) under four scenarios of species' flexibility (solid to dotted lines) to reallocate interactions to persisting partners (constrained rewiring); here secondary extinction is triggered after 50% interaction loss. In this network, sensitivity to plant extinction (red area) was larger than sensitivity to animal extinction (blue area), that is, animal species went more quickly secondarily extinct than plant species. Secondary extinction plots for the 12 other interaction networks are shown in Supplementary Fig. 1.

studied networks comprised more animal than plant species (Supplementary Data 1) and are thus better buffered against animal than plant extinction[11]. However, differences between secondary animal and secondary plant extinction were generally larger under climate-induced extinction than under random extinction. As random extinction accounts for differences in animal and plant species numbers and mean specialization in each network, differences in species coextinction between climate change and random extinction must be due to an alternative second mechanism. Different impacts of plant and animal extinction on the networks are, thus, linked to the different relationships between biotic specialization and vulnerability to climate change for plants and animals. As animal species with the highest vulnerability to climate change had a low diversity of plant partners (Fig. 1c,d), the loss of these animal species had a weak impact on the networks and did not disrupt the backbone of interactions in the ecological community (Fig. 2c).

Our results suggest that animal extinction under climate change will only weakly affect animals' ecological function to plants, such as pollination and seed dispersal. Although this finding has important implications for ecosystem functioning, the simulations did not account for variability in the functional quality of different animal mutualists[24]. Thus, the inference of our simulation model is limited to quantitative contributions of animals to ecosystem functions[18], such as the number of visits by animal pollinators or seed dispersers. Nevertheless, our simulation model suggests a high robustness of plants to animal extinction in future communities. This is consistent with the finding that mutualistic networks on islands generally lack a high diversity of animal pollinators and seed dispersers, but apparently maintain their pivotal functions to plants[28,29]. In natural communities, the tolerance of plants to the loss of their mutualistic animal partners is further increased because many plant taxa can locally persist for extended time periods[25]. This suggests that the functional dependence of plants on their animal pollinators and seed dispersers is comparatively low and, at least for some plant taxa, may be weakly related to the reciprocal dependences derived from mutualistic plant–animal networks.

In contrast to the robustness of plants to animal extinction, our simulation model shows that plant extinctions could trigger

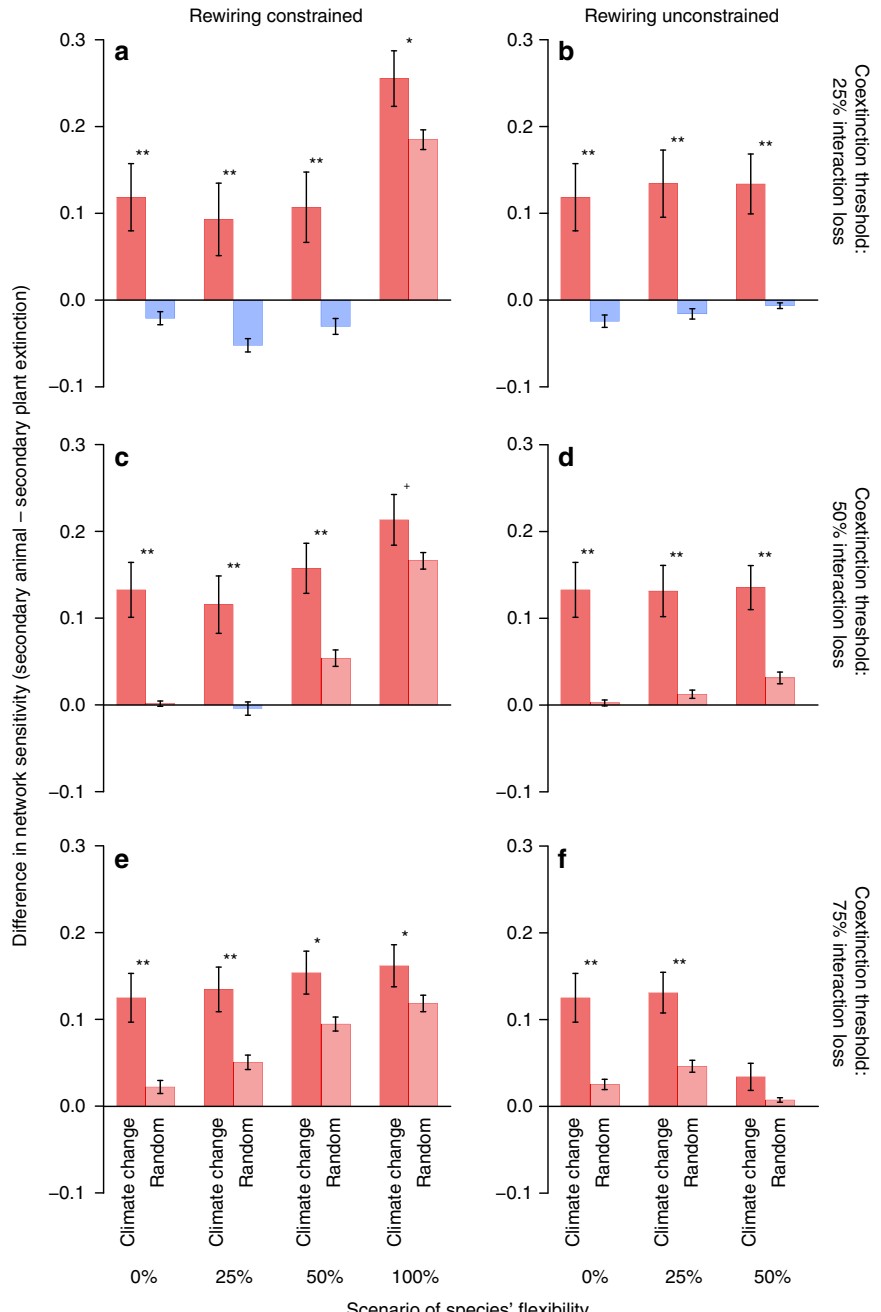

**Figure 3 | Differences in sensitivity to species extinction across 13 mutualistic networks.** Shown are differences in network sensitivity to plant versus animal extinction for different scenarios of species' sensitivity to coextinction, rewiring capacity and flexibility. Coextinction thresholds varied between (**a,b**) 25%, (**c,d**) 50% and (**e,f**) 75% of interaction loss. Species were able to rewire interactions (**a,c,e**) to persisting partners (constrained rewiring) or (**b,d,f**) to all persisting species in each network (unconstrained rewiring). Flexibility values (0%, 25%, 50%, 100%) indicate the proportion of lost interactions that was reallocated to other species in the respective scenario; we omitted the very unlikely scenario of unconstrained rewiring and 100% flexibility as it requires all species to go extinct to trigger secondary extinction. Shown are mean differences (±1 s.e.) across the 13 pollination and seed-dispersal networks between the impact of plant versus animal extinction; values >0 (red bars) indicate a higher risk of secondary animal than secondary plant extinction and values <0 (blue bars) indicate the opposite. Secondary animal versus secondary plant extinction was compared between climate change and random extinction using two-sided, pair-wise t-tests (+P<0.1; *P<0.05; **P<0.01). Here climatic projections of the models of species' vulnerability to climate change follow the RCP 8.5 scenario; results were identical for the RCP 6.0 scenario (see Supplementary Fig. 2).

bottom-up trophic cascades[13,26,27]. We found a high risk of animal coextinction, especially in simulations where animals were rather inflexible in choosing their plant partners. Apparently, the higher sensitivity of mutualistic networks to plant than to animal extinction disappeared only in simulations where species could freely reallocate at least 50% of their lost interactions to all

persisting species in the network (Fig. 3f). Empirical studies have shown a high variability of mutualistic plant–animal interactions across years[30,31], suggesting a high flexibility in these networks. In contrast, recent studies have highlighted the importance of partner fidelity in both antagonistic and mutualistic ecological networks[32,33] that might be associated with a high degree of trait

matching between interacting species in these networks[34,35]. According to these studies, the flexibility of interactions will be constrained in ecological networks under future conditions, suggesting that unconstrained rewiring of interactions is unlikely even for generalized mutualistic networks. This notion is corroborated by a substantial loss of interactions in a plant–pollinator network during the past century, although rewiring of interactions mitigated the loss of pollinator function in the novel community[36].

Under the assumptions that interaction frequencies are surrogates of functional dependences between species[18] and that no new species will enter the networks, we show that plant extinctions in response to climate change will cause cascading effects on animal species in mutualistic networks. Such bottom-up effects may be particularly severe for animal groups that depend on plant resources throughout their life cycle (that is, in the larval and adult stage) and are restricted to specific resource types, such as many insect taxa[7,23], rather than for animals that are able to use alternative resources, such as most birds[37]. Indeed, a high risk of extinction cascades from lower to higher trophic levels has been shown for specialized plant–insect interactions[13]. In addition, direct effects of habitat and climate change on animals are likely to exacerbate their indirect effects mediated by bottom-up extinction cascades[13]. To quantify the adaptive capacity of animals and plants in ecological networks, long-term studies across multiple communities or translocation experiments for specific communities[38,39] will be needed. Such experiments will also be useful to test whether the intensity of the indirect effects of climate change on biodiversity varies among functional groups of plants and animals.

Biotic interactions have often been neglected in assessments of the impacts of climate change on biodiversity[3,4]. Here we demonstrate for central Europe that animal pollinators and seed dispersers that interact with a low diversity of plant partners are particularly vulnerable to climate change. In contrast, plants' vulnerability and biotic specialization are unrelated. This difference between animals and plants has important consequences for their projected coextinction risks under climate change as cascading effects from plants to animals are likely to trigger animal coextinctions in mutualistic networks and can aggravate impacts of climate change on biodiversity. Accounting for biotic interactions between species is, therefore, important for accurately predicting impacts of climate change on animals, whereas plant–animal interactions are less relevant for predicting plant responses to climate change. Correspondingly, ecological functions to plants, such as pollination and seed dispersal by animals, appear to be robust to climate-induced extinctions of animals.

## Methods

**Plant and animal occurrence data.** We compiled occurrence data for plant and animal species recorded in eight quantitative pollination and five quantitative seed-dispersal networks from central Europe (Supplementary Data 1). Range maps of plant distributions were compiled from published distribution maps[40–42], occurrence data from the Global Biodiversity Information Facility (GBIF), national and regional floristic databases and further maps from the floristic literature (see bibliographic details in Index holmiensis[43–46]). Contiguous large areas of plant occurrence were generalized as range polygons; spatially isolated occurrences were digitized as single-point locations. Distributional data on most wild bee taxa were extracted from GBIF and from a database hosted at the University of Mons (Belgium) (Atlas Hymenoptera[47–53]). Most of the *Andrena* bee data were originally derived from maps associated with the Warncke collection (Biocentre of the Upper Austrian Museum, Linz, Austria[54]). Data on distribution of *Colletes* bees were provided by Michael Kuhlmann. Distributional data for European butterflies were provided by the database LepiDiv[55], an updated version of the database used for the Distribution Atlas of Butterflies in Europe[56]. Data for hoverflies were provided by a database hosted at the University of Novi Sad[57]. Range maps of bird distributions were compiled from a database of global avian distribution maps[58]. All distribution data were gridded to match a European CGRS grid (3024

equal-area cells with a resolution of 50 × 50 km). We omitted areas from eastern Europe with a low sampling intensity for insect pollinators (see Supplementary Fig. 3 for the spatial extent of the used grid). Data aggregation at the 50 × 50 km grid further mitigates effects of low sampling intensity, that is, an overestimation of species' absences for some insect taxa.

**Climatic niche estimation and species distribution models.** We computed the current range size of a species as the number of grid cells with observed presences within the European CGRS grid; in the case of avian migrants, only presences within the breeding range were considered. We omitted species from the analyses for which no or deficient occurrence data were digitally available; the number of occupied grid cells per species ranged from 19 to 2,917 (median = 815 occurrences across the 50 × 50 km grid, Supplementary Data 2). Species' realized climatic niches were quantified as a function of their occurrences and four variables of current climate[59] (annual mean temperature, temperature annual range, annual precipitation and precipitation seasonality, sampled for the used grid). We used two alternative methods for quantifying the current climatic niche breadth of a species (see Supplementary Fig. 4 for a data overview). First, we used the hypervolume method[60], which calculates the realized climatic niche breadth by using a multidimensional kernel density estimation procedure to estimate an $n$-dimensional hypervolume from a set of species' occurrences and the respective climatic variables. To calculate the climatic hypervolume, we $z$-standardized climatic variables and used a Silverman bandwidth estimator and a 0% quantile threshold[60]. Second, we used the outlying mean index (OMI) that quantifies the distance between realized and background climate conditions[61]. In contrast to other multivariate methods for niche quantification (such as canonical correspondence or redundancy analysis), it makes no assumption about the shape of the response curve to the environment (that is, climate) and is not influenced by species richness[62]. Along each ordination axis, the OMI calculates niche breadth as variances based on the climatic conditions at the localities of species' occurrences. Based on the first and second OMI ordination axes (cumulative inertia of both axes: 89%), we defined species' realized niche breadth as the geometric mean of variances along these two axes. Both methods resulted in similar estimates of realized climatic niche breadth ($n = 709$ species, $r = 0.92$) and were positively associated with the current range size of a species (hypervolume, $r = 0.54$; OMI niche breadth, $r = 0.67$). We also tested whether the spatial extent of the analysis may have affected estimates of climatic niche breadth. We found that the OMI climatic niche breadth, derived from occurrences across the entire Palearctic for plants and birds, was closely correlated to that at the European scale ($n = 346$ species, $r = 0.83$). Occurrence data beyond Europe were not available for insect pollinators and all taxa were therefore analysed at the European scale.

We quantified probabilities of occurrence from species' recorded presences and absences with species distribution models based on boosted regression trees[63], using a cross-validation approach to estimate the optimal number of trees (number of initial trees = 10, tree complexity = 2, learning rate = 0.01). To evaluate model performance, we calculated the area under the receiver-operator curve (AUC) and the True Skill Statistic (TSS)[64] where the sum of the model's sensitivity and specificity was maximal ($TSS_{max}$). AUC and $TSS_{max}$ were calculated as the arithmetic mean of 10 random splits of data into 75% used for model calibration and 25% for model testing. Arithmetic means ( ± 1 s.d.) of $AUC/TSS_{max}$ values were: $0.85 ± 0.046/0.58 ± 0.093$ (bees), $0.84 ± 0.057/0.57 ± 0.120$ (butterflies), $0.79 ± 0.057/0.49 ± 0.107$ (hoverflies), $0.94 ± 0.031/0.77 ± 0.074$ (birds), and $0.95 ± 0.026/0.80 ± 0.073$ (plants), indicating good to very good (pollinators) or excellent (birds, plants) model performance under current conditions (Supplementary Data 2). We used the full set of current occurrences of each species for calculating model projections under current and future conditions. Future climate projections were obtained from two general circulation models from the Fifth Assessment Report of the Intergovernmental Panel on Climate Change (IPCC) (CCSM4, MIROC5 (refs 59,65)) using two scenarios of RCPs assuming an average increase of $2.85 ± 0.62 °C$ (RCP 6.0) or $4.02 ± 0.80 °C$ (RCP 8.5) in mean annual temperature for the geographic area covered (see Supplementary Fig. 3 for the projected changes in all climatic variables). We quantified the potential vulnerability of a species to projected climate change as changes in climatic suitability for each grid cell covered by the species' current European range, defined by the difference between the probabilities of occurrence under current (years 1950–2000) versus projected future conditions for 2070 (averaged for 2061–2080). For each species, changes in climatic suitability were summarized using the median change across all grid cells of the species' current European range; projections were averaged between the two circulation models and calculated separately for each RCP scenario (see Supplementary Fig. 4 for a data overview). Restricting the vulnerability quantification to a species' current distribution may overestimate extinction risk because areas outside the current range, which may become suitable in the future, are not considered. However, this approach accurately quantifies species' exposure to projected changes in climatic conditions and avoids several simplifying assumptions, for example, about species' dispersal ability, non-analogue climates or novel biotic interactions that are particularly problematic in range projections beyond current distributions[17].

Changes in climatic suitability were very closely correlated between the two RCP scenarios ($n = 709$ species, $r = 0.97$), but were only weakly related to current climatic niche breadth ($n = 709$ species, climatic hypervolume, $r < 0.5$ for both RCP

scenarios; OMI climatic niche breadth, $r < 0.4$ for both RCP scenarios). Changes in climatic suitability at the regional scale (that is, within the nine grid cells adjacent to each network's study region) were closely correlated to range-wide suitability changes ($n = 709$ species, $r = 0.74$ for both RCP scenarios). We used range-wide suitability changes because they are more representative for a species' vulnerability across its range and are less sensitive to local climatic projections.

**Mutualistic pollination and seed-dispersal networks.** We matched data on climatic niche breadth and vulnerability with empirical data of biotic specialization derived from eight quantitative pollination and five quantitative seed-dispersal networks, each recorded within a different region in central Europe (Supplementary Data 1). We included these networks in the analyses because they report comprehensive data on interaction frequencies between species pairs, consistently recorded by direct plant observations over several months (Supplementary Data 1). Interaction frequencies equal the number of visits of an animal to a plant species. To cover all potential interaction partners of a species within the regional context, we summed interaction frequencies over time and space within each region as most networks were recorded on repeated visits at several localities within each region. Interaction data from other animal taxa than bees, butterflies, hoverflies and birds were excluded from the original networks. As it is generally the case for mutualistic networks, we lack information on the actual functional dependences of pollinators and seed dispersers on their foraging plants and vice versa and, thus, assume that interaction frequency is closely associated with the reciprocal functional importance of the interacting species[18]. Overall, the 13 pollination and seed-dispersal networks were characterized by a low to intermediate degree of specialization; complementary specialization ($H_2'$) ranged from 0.29 to 0.47 (mean = 0.38, see Supplementary Data 1 for a compilation of other standard network metrics for the 13 mutualistic networks).

In total, we were able to derive independent data of biotic specialization, climatic niche breadth and vulnerability to climate change for 363 insect (196 bees, 70 butterflies, 97 hoverflies; pollinators hereafter), 51 bird (seed dispersers hereafter) and 295 plant species (Supplementary Data 2). These species comprised most of the species of the respective study taxa (bees, butterflies, hoverflies, birds and plants) in the original networks (mean across networks: 88%; range: 74–100%, see Supplementary Data 1). Including species with deficient occurrence data and/or other animal taxa for which no distribution data were available (for example, other Diptera or Coleoptera that had been sampled for a few of the original pollination networks) resulted in qualitatively identical estimates of biotic specialization for the analysed plant and animal species.

**Network analysis and simulations of species coextinctions.** We measured plant and animal specialization in two ways based on the number and uniqueness of interaction partners within each network. First, we calculated the effective number of partners. It equals the diversity $e^H$ of interaction frequencies per species (based on the Shannon index $H$) and is equivalent to the number of partners if each link was equally common[66]. This metric is positively related to the total interaction frequency of plants (log–log scale, $n = 591$, $r = 0.68$) and animals ($n = 1009$, $r = 0.76$). On average, plants had a higher diversity of partners than animals (mean $e^H \pm 1$ s.e., $4.20 \pm 0.17$ versus $2.61 \pm 0.10$). Second, we calculated complementary specialization $d'$ as the deviation between the observed interactions and partner selection according to species' total interaction frequencies[67]. The metric ranges from 0 for a generalist species (sharing partners with many others) to 1 for a fully specialized species (with a unique set of partners). $d'$ was unrelated to total interaction frequency (plants, $r = 0.03$; animals, $r = 0.06$) and was similar for plants and animals ($0.34 \pm 0.01$ versus $0.32 \pm 0.01$).

We tested the statistical associations between species' biotic specialization within each network and climatic hypervolume (square-root transformed), OMI climatic niche breadth (geometric mean of variances along the first two OMI ordination axes) and projected changes in climatic suitability (median change across a species' current European range) with linear mixed-effects models; error distributions of all models did not deviate from normality. We fitted statistical models including main and interaction effects of the respective climatic predictor and trophic level (animal versus plant) on biotic specialization. In all models, we accounted for random variation due to network identity, plant and animal taxonomy on the model intercepts (taxonomic levels: family, genus, species). To account for variation in the performance of species distribution models and, thus, for the uncertainty of projected changes in climatic suitability, we weighted the respective linear mixed-effects models with the accuracy of the respective species distribution model as given by the $TSS_{max}$ value[64] for each species (Supplementary Data 2). In the interest of comparability, we z-transformed realized climatic niche breadth (climatic hypervolume, OMI climatic niche breadth) and changes in climatic suitability before the statistical analyses. In order to examine network-specific relationships between biotic specialization and climatic variables, we additionally tested effects of climatic predictors on biotic specialization in models accounting for both random intercepts and slopes of network identity, separately for plants and animals.

Based on the projected impacts of climate change on plant and animal species, we modelled effects of climate change on each network. We simulated secondary species extinction as a consequence of the sequential loss of plant and animal species, respectively[11,12]. The order of species loss followed the projected changes

in climatic suitability; thus, we first removed the species experiencing the largest decline in climatic suitability across the current European range, followed by the removal of the species with the second largest decline until the least vulnerable species had been removed. In the simulation model, species became secondarily extinct once they had lost at least 25, 50 or 75% of their interaction events in respect to the original network, assuming that interaction frequencies are proportional to the functional dependences of animals on plants and vice versa[18]. These thresholds for secondary extinction are probably more realistic than the assumption that all interaction events have to be lost before a species goes extinct[15]. We further assumed that species could reallocate lost interactions to other persisting species in the network accounting for the flexibility of partner choice in interaction networks[13,16]. We simulated two different rewiring scenarios under the assumption that no new species will enter the network. First, we reallocated a varying proportion of removed interactions to all persisting partners (constrained rewiring), proportional to the relative interaction frequencies of each species. Second, we reallocated lost interactions to all persisting species (unconstrained rewiring), relative to species' total interaction frequencies. Thus, the first scenario assumes that species will be constrained in their interactions to their current partners, whereas the second scenario assumes that species are able to freely establish new links under future conditions. We varied the flexibility of species to reallocate lost interactions to persisting species between 0% (no reallocation) and 100% (reallocation of all interactions). In the scenario where all partners are interchangeable (100% flexibility), all interaction events must be lost to trigger secondary extinction as all lost interaction events are reallocated to persisting species. In a scenario of unconstrained rewiring and 100% flexibility, thus, all species need to go extinct to trigger secondary extinction. As this is a very unlikely scenario, it was omitted from our simulations.

For each network and simulation scenario, we quantified network sensitivity to plant and animal species extinction by the area above the secondary extinction curve; the metric ranges from 0 (no species go secondarily extinct) to 1 (all species go secondarily extinct after removing a single species) and is conversely equivalent to network robustness (the area under the extinction curve[12]). We computed the difference in network sensitivity to plant and animal extinction for each RCP projection and each coextinction, rewiring and species' flexibility scenario. We further compared network sensitivity for the different simulation scenarios between projected climate change and 200 random sequences of species loss; here network sensitivity was averaged across iterations of random extinction for each simulation scenario. We tested whether the risk of secondary animal versus secondary plant extinction differed between climate change and random extinction using two-sided, pair-wise $t$-tests.

**Data availability.** Estimates of biotic specialization (effective partners[66], $d$[67]), range size (number of occupied grid cells), climatic niche breadth (climatic hypervolume[60], OMI climatic niche breadth[61]) and vulnerability to projected climate change (RCP 6.0 and RCP 8.5 scenarios[65] plus model accuracy estimated by AUC and $TSS_{max}$ values[64]) are reported for all 295 plant, 196 bee, 70 butterfly, 97 hoverfly and 51 bird species in Supplementary Data 2. The pollination and seed-dispersal interaction network matrices are available from the authors on reasonable request (see Supplementary Data 1 for metadata). We used code from the following freely available R packages for the statistical models and simulations: 'hypervolume', version 1.4.1 (ref. 60), 'ade4', version 1.7.4 (ref. 68), 'gbm', version 2.1.1 (ref. 63), 'bipartite', version 2.06.1 (ref. 69) and 'lme4', version 1.1.12 (ref. 70).

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

## Acknowledgements

We thank Aidin Niamir for compiling data on current and future climates. M.S., K.B.-G., D.M.D. and T.H. received support from the research funding programme 'LOEWE–Landes-Offensive zur Entwicklung Wissenschaftlich-ökonomischer Exzellenz' of Hesse's Ministry of Higher Education, Research and the Arts. J.F. was supported by a DFG research fellowship (FR3364/1-1). O.S., D.M., P.R., J.S., M.W. and A.V. received funds from FP7-STEP 'Status and Trends of European Pollinators' project. O.S. and D.M. also acknowledge support from the EU COST Action Super-B. J.A. received funds from the German Academic exchange service. M.B., N.E., A.K. and A.S. received funds from DBU (German Federal Environmental Foundation) and the German Federal Ministry of Education and Research. G.B. received funding through the Bavarian research cooperation 'Climatic Impacts on Ecosystems and Climatic Adaptation Strategies'

(FORKAST). N.B. and C.N.W. received funds from the DFG Priority Program 1374 'Infrastructure-Biodiversity-Exploratories'. D.M. and P.R. received funds from FNRS (Belgium) 'Web-impact' and from BELSPO 'Belbees' projects.

## Author contributions

M.S., J.F., O.S., N.B., K.B.-G., D.M.D., C.F.D., T.H., I.K. and C.H. conceived general ideas. O.S., E.W., H.B., A.H., M.K., I.K., D.M., S.M.-S., P.R., J.S., A.V., M.W. and C.H. contributed distribution data. M.S., J.F., J.A., M.A., M.B., G.B., N.B., N.E., N.F., A.K., M.P., A.S. and C.N.W. contributed network data. M.S., J.F., O.S., E.W. and C.H. analysed the data. M.S. and C.H. wrote the manuscript. All authors discussed the results and commented on the manuscript.

## Additional information

**Competing financial interests:** The authors declare no competing financial interests.

