## [Peer Review File · Nature Communications]

Reviewers' Comments:

Reviewer #1 (Remarks to the Author):

Abstract: The result that plant extinctions are more likely to trigger animal extinctions that vice versa is counterintuitive, given the well documented extinctions of animals on islands, but very few documented extinctions of the plants that rely on them for dispersal. So, this is interesting but may require more discussion.

The data and methods seem carefully thought out, but I was surprised to see an infrequently-used method of estimating species climatic niches, the OMI. I am unfamiliar with this method and I am not sure how it compares to other methods. It may be a perfectly rigorous and appropriate method, but I would like to see more explanation and justification.

Other methods:

Blonder B, Lamanna C, Violle C, Enquist BJ. 2014. The n-dimensional hypervolume. *Global Ecology and Biogeography* 23:595-609.

Broennimann, O., Fitzpatrick, M. C., Pearman, P. B., Petitpierre, B., Pellissier, L., Yoccoz, N. G., Thuiller, W., Fortin, M.-J., Randin, C., Zimmermann, N. E., Graham, C. H. & Guisan, A. (2012) Measuring ecological niche overlap from occurrence and spatial environmental data. *Global Ecology and Biogeography*, 21, 481-497

Petitpierre, B., Kueffer, C., Broennimann, O., Randin, C., Daehler, C. & Guisan, A. (2012) Climatic niche shifts are rare among terrestrial plant invaders. *Science*, 335, 1344-8.

Species distribution models:

What is the sample size for each model? Were absences or pseudo-absences used? How were they selected? How was bias in the sample addressed in the modeling? This can have a big effect on SDMs.

AUC for some groups actually seems a bit low (<0.9) although the literature would classify this as 'good.' Also AUC based on presence versus pseudo-absence has been criticized - were other metrics considered (TSS)?

Although the range size and niche breadth are calculated from atlas data, the change in climate suitability for grid cells is calculated from the modeled current climatic suitability and projected future climatic suitability. So, any discrepancy between current distribution and current modeled distribution is going to add noise to the analysis of vulnerability. Does this require a high degree of accuracy of the SDMs? Can you comment on how this uncertainty affects the conclusions?

The SDM-based projections of species' vulnerability to climate change (change in climatic suitability) are really estimates of species exposure to climate change. As pointed out by Schwartz, vulnerability depends on characteristics of the species (niche breadth and biotic interaction among them), and this approach tends to overestimate species' vulnerability to extirpation.

Schwartz MW. 2012. Using niche models with climate projections to inform conservation management decisions. *Biological Conservation* 155:149-156.

Can you discuss the effect of the coarse scale (50x50 km) of the analysis? Species that occur together within a 50x50km grid cell may not actually interact. Furthermore, projected changes in climatically suitable habitat are sensitive to data resolution (Franklin et al. 2013). So the co-occurrences in the analysis may be an artifact of scale. How might this affect your conclusions?

FRANKLIN, J., DAVIS, F. W., IKAGAMI, M., SYPHARD, A. D., FLINT, A., FLINT, L. & HANNAH, L. 2013. Modeling plant species distributions under future climates: how fine-scale do climate models need to be? *Global Change Biology* 19(2):473-483.

What is d' in Table S4?

Reviewer #2 (Remarks to the Author):

In the manuscript "Ecological networks are more sensitive to plant than animal extinction under climate change" Schleuning et al. present a novel way (at least to me) of integrating climate change and species interactions. In particular, they show that animal species with a more specialized interaction nature tend to have smaller geographical ranges, tend to have smaller climatic niches and tend to be the ones most severely affected by climate changes. A similar pattern was not seen for plant species. Furthermore, they look into how primary removal of animals and plants may result in secondary extinctions and conclude that plant removal is more severe than animal removal. To me the former finding (i.e. the relationship between biotic specialization and geographical ranges, climatic niches and climatic vulnerability) seem more interesting because the later could (as far as I can tell, but I might be mistaking) be strongly

influenced by the fact that plants tend to have more interaction partners than animals (see also a more detailed comment below). I find the manuscript of a high quality, well written, and very interesting and I think it could be of broad interest for the ecological community. The statistical analyses seem valid and the conclusions seem robust, with appropriate credit to previous work. That said, I think it could benefit from more precise and detailed descriptions at several occasions. See specific comments below.

I'm missing a clear explanation of how the authors selected which interaction networks to include in their dataset. Which criteria were used? Without such justification it may seem as a random sample (maybe based on accessibility) with a strong preference for networks located in Germany.

Regarding the interaction network it is not completely clear to me how many species from the original networks that were omitted. In e.g. Weiner et al. (2014) (used three times in the dataset) interaction network consisted of Diptera, Hymenoptera, Lepidoptera and Coleoptera. However, in the current analysis the authors have restricted their focus to Syrphids (one family within Diptera), Hymenoptera and Lepidoptera.

The authors provide some numbers in table S1 that states how many animals that were removed, but do these numbers include all removed species or only removed species belonging to Syrphids, Hymenoptera and Lepidoptera? Some more precision is needed here.

In a similar vein, in line 237-238 the authors write: 'Including species with deficient occurrence data in the calculation of network metrics resulted in qualitatively identical results'. Therefore, when the authors write 'species with insufficient occurrence data' are they then referring to all animals originally included in the network or only Syrphids, Hymenoptera and Lepidoptera with insufficient occurrence data? I think it is important to state if these calculations were done on a subset of the original data or not? Therefore, some more details are needed here.

Regarding the discussion of why the communities are more sensitive to animal removal than plant removal I think that the authors second mechanism (line 119-124) is probably the primary cause. That is, the reason why communities are more sensitive to plant extinction compared to animal extinction may lie in the differences in linkage level (a difference that is partly documented in line 252-253, and which also can be seen from figure 1). Assuming that most plant species likely have more interaction partners than animal species (and on average receive more total number of visits), and that more animals than plants rely on a single (or a few) plant species (at least that seems plausible from the networks in figure 2), it can be expected that plant removal will inevitably result in more animals experiencing secondary extinctions, than the removal of animals will result in plant secondary extinctions (which I think is also what figure 3 demonstrates when the flexibility is set to 100%). I think this needs to be highlighted and discussed more explicitly around line 119-124.

On a more general note I have a hard time understanding why the 'climate change' and the 'random' removal scenarios become more equal when species flexibility increases. Indeed this difference is not at all discussed in the paper and should be included somewhere. If I understand it correctly the 100% flexibility is equal to a situation where species only goes extinct when they lose all their interaction partners, and as such this overrules the authors initial criteria that species goes extinct once they lose 50% of their interactions. I'm not arguing against this 50% reduction methodology, but the results and I particular the contrasting results of having 0% vs. 100% flexibility, and why it becomes closer to random removals, should be discussed more deeply.

Line 6: Missing comma after species?

Line 16: What is the difference between 'tolerate' and 'adapt'. Is the former implying no response - in the sense that species are unaffected by the alterations? If so, the sentence "Climate change forces species...to tolerate..." sounds a bit wrong. Moreover, I think this first sentence needs a bit more work to increase readability.

Line 31-33: Would it make more sense to split it into three distinct test/hypothesis (i) Small realized ranges, (ii) narrow climatic niches and (iii) projected loss in climatic suitability. These tests are also treated separately in the results.

Line 36: I think it would be more informative if the authors specified how many plants, insect pollinators and avian seed disperses, respectively, they have included (as specified in the Materials and Methods), and not have many they have in total across all three categories. It would also be more precise to state these numbers in the abstract (line 5).

Line 41: The authors needs to specify what exactly 'interaction events' covers. I would suggest using 'visitation events' as it could otherwise be incorrectly confused with number of unique interactions. Alternatively, the authors could more precisely specify what they mean.

Line 49: If I understand the Materials and Methods correctly it really isn't the number of animal partners that the authors are looking at. Maybe it would be more accurate to write what it actually is, viz. 'diversity in partners' (measured with Shannon-Weaver,). Or, at least call it effective partners as done in line 47. Actually, the authors should be more coherent throughout and for example use 'effective numbers of partners' when referring to this metric.

Line 49-51: I guess a similar statement could be made for 'climatic niche breadth'?

Line 72: Should it be 'effective partners'.

Line 119: I suggest that the authors start the sentence with 'Second', so that it follows the logic of presenting the two different mechanisms.

Line 125-126: I'm not sure I understand this sentence. Which ecological function are the authors refereeing to, and which results are supporting this?

Line 143: Delete 'a'?

Line 168: Maybe the authors should add that all networks were quantitative (although this becomes apparent further down in the Materials and Methods). The authors might also add this information earlier in the manuscript.

Line 248-250: Based on the description I'm a bit unsure whether the metric 'effective number of partners' is defined exactly as e^H , or if it is some other measure based on e^H which can be found in Bersier et al. 2002?

Line 252-253: Strictly speaking, when using e^H it is not possible to say that plants interacted with more partners than animals, as compared to just interacting more equally with the partners they actually have. I suggest rephrasing to something like: '...plants had a higher diversity of partners....'

Line 254: Should 'a' be removed?

Line 259-265: Did the authors allow both random intercept and slope in their linier mixed model as suggested by for example Schielzeth & Forstmeier (2009). Please explain.

Line 266-269: I'm not completely sure what the authors have tested here, and how it deviates from the linier mixed model procedure described in line 259-265. Please provide some more details.

Figure 2

In the figure legend the authors referrer to 'light shaded' and 'dark shaded' as a scale that goes from high to low vulnerability. However, in the figure it is the interaction that are coloured end not the species, which is a bit confusing. Although this colouring of the interactions could be meaningful I would suggest arranging the species from low to high vulnerability. That would make the removal sequence much more apparent at first sight, and would likely make it more convenient to relate the colour of the interactions with the scale-bar at the bottom.

Furthermore, I would suggest renaming the labels of the sensitivity plots so that on the x-axis it would be 'plants removed' or 'animals removed', and on the y-axis it would be 'animals persisting' or 'plants persisting'. That would greatly aid readers not familiar with such plots. Finally, the authors could even consider labelling the two rows of the bipartite plots with 'animals' and 'plants'.

References:

- Bersier, L.-F., Banašek-Richter, C. & Cattin, M.-F. (2002) Quantitative descriptors of foodweb matrices. *Ecology*, 83, 2394-2407.
- Schiellzeth, H. & Forstmeier, W. (2009) Conclusions beyond support: overconfident estimates in mixed models. *Behavioral Ecology*, 20, 416-420.
- Weiner, C.N., Werner, M., Linsenmair, K.E. & Blüthgen, N. (2014) Land-use impacts on plant-pollinator networks: interaction strength and specialization predict pollinator declines. *Ecology*, 95, 466-474.

Reviewer #3 (Remarks to the Author):

Effects on climate change on individual species have been documented in many cases. More recently, there has been a growing interest in the effect of climate changes on interactions among species. Iconic examples are phenological mismatches between herbivores and plants, or predators and prey. However, most of these studies focused on a few case examples, and there are only few studies that looked at effects of climate change (or other environmental changes) on large interaction networks (e.g. Peasre et al 2013).

In their manuscript, Schleuning et al. addressed how interaction networks (using empirical data on plant-pollinator and plant-seed-dispersal networks) could be affected by climate change. They find in a combination of species-distribution models and ecological network analysis that different groups of species and their interactions are differently vulnerable to be affected by climate change. The main finding is that animal species that are specialized on few plants are more vulnerable to climate change, while biotic specialization of plants is not related to vulnerability in the context of climate change. Consequently, animal species are supposed to be more likely affected by co-extinctions than plant species.

This finding is interesting, and has to my knowledge not been shown in such a comparison of several interaction networks. However, the finding is by itself not very surprising, see for example classic work by Warren et al. 2001. The models used in the study make the relatively strong assumption that the interaction-link strengths are all equal, and-even stronger assumption-that links between different functional groups are directly comparable. I think this is clearly not

the case, and should at least be stated. The finding that removing links has stronger effects on animals depending on plants than vice-versa depends on the assumption that the significance of the links is a priori the same (key finding line 108-109). I am not sure if that can indeed be assumed; at least it is a very strong assumption and this limitation should be better discussed.

While the text reads generally well, I found it a bit narrative and I was hoping to see more of the underlying support/link to the data to better support the conclusions. I do not say this support is not there, just that it should be much more specifically shown. For example, data points in the figures (e.g. Fig. 1) should be added, not only model fits. Also, much more details of the 13 interaction networks could be given. It would be helpful to have a visual depiction of all individual networks, even in the main article, and a summary table of the key network parameters generally used in mutualistic networks (e.g., nestedness, modularity, number of links, etc.). For example, the findings reported for one interaction matrix (Fig. 2) should be given for all 11 remaining ones as well (supplement); also, it is nowadays standard that the interaction matrices are provided in a machine-readable form.

The interaction strengths but also limitations of the findings with respect to the individual interaction matrices should be discussed.

Specific additional comments:

- There is already quite a bit of literature that has been analyzing co-extinctions in plant-insect interaction networks, looking for example at how range size etc. are affecting the extinction cascades (e.g., Peasre et al. 2013 Ecology; this and other work might be worth to be cited as it is using similar approaches). Also, one of the other main findings, namely that specialized animal species may be more vulnerable to climate change than generalists, is not novel (e.g., Warren et al. 2001 Nature). These findings should be better discussed in the context of previous work.
- It was not clear to me if the observed correlation between realized range size and effective number of interaction partners referred to effective number of local interaction partners, or across the range (lines 46 to 48). It would be possible to have few interaction partners locally, but many regionally (i.e., different partners at different localities) or to have many partners locally, but everywhere the same. The consequences with respect to extinction cascades are very different: in the former case a few losses are dramatic, while in the latter the networks are highly buffered. Please clarify.
- The study looked at several (13) interaction networks, which is laudable. However, they are often treated as one ("720 plant and animal species in pollination and seed-dispersal networks"). I think it would be better to refer more to the actual number of networks, and their structure, than

on the total number of species only (e.g., abstract). Especially in the main text these networks are a bit treated as a "black box", and the discussion of the results is often at a meta-level, where details to the individual networks is lacking. Also, I missed more detailed information and description of these networks. It would be relatively easy to visualize these 13 networks. Also, the limitations of these interaction networks should be clearly discussed: do I understand it correctly that interactions were inferred from co-occurrences in 50x50 plots? This is a rather strong assumption for organisms with an often much smaller mobility.

- The study makes the strong assumption that a loss of interaction in a pollinator-plant network causes subsequent extinctions. Is this realistic? I think this may be realistic for some bee species which are relatively specialized pollinators, but much less so for butterflies. This aspects needs much better justification. Butterfly pollinators are notoriously unspecific in their choice of nectaring plants, and to my knowledge there is relatively few plant species in Central Europe that are depending on a few key pollinators, and very few or even none of them is a "butterfly plant". Thus, it is empirically not very strongly supported assumption that loss of pollination interactions results in a loss of species. This at least needs to be discussed, or different levels of interaction strengths could be introduced in the model.

- Some of the studied species (especially butterflies) are not only having mutualistic interactions with plants, but also antagonistic interactions in their larval stage. While nectaring/pollinating is an important interaction, it is likely much less essential for a butterflies' persistence than the antagonistic interaction in the larval stage, and is arguable also much less specific than for example interactions of bees, which are known to me much more host-specific pollinators, while butterflies are more generalist pollinators.

- The study assumes that interactions are fixed and can't be "rewired" (lines 100-103). While I can see that this assumption makes simplifies the data analysis, I am not convinced that it actually reflects reality: in the context of climate change novel interaction partners show up, and plant-insect networks have lately been shown to be rather flexible in incorporating novel trophic interactions. Given that the study is based on models and extinction scenarios, it would be relatively easy to incorporate also possible host switches, and do so under different assumptions. I would strongly recommend to do additional work showing how robust/dependent the results are on assumptions regarding host-use flexibility.

Literature mentioned:

Pearse et al. 2013 Extinction cascades partially estimate herbivore losses in a complete Lepidoptera-plant food web. *Ecology* 94, 1785-1794.

Warren et al. 2001 Rapid responses of British butterflies to opposing forces of climate and habitat change. *Nature* 414, 65-69.

Point-by-point response to reviewers

We thank the referees for their thorough reviews and valuable suggestions. In the revised manuscript, we have accounted for all comments, particularly by adding numerous analyses demonstrating the robustness of our findings. We provide an itemized response to each comment below; our responses to the referees' comments are written in italics.

We hope that the referees agree that the revised manuscript has improved substantially and that it now may be acceptable for publication.

*Sincerely,
Matthias Schleuning and Christian Hof (on behalf of all authors)*

Reviewer #1 (Remarks to the Author):

Abstract: The result that plant extinctions are more likely to trigger animal extinctions that vice versa is counterintuitive, given the well documented extinctions of animals on islands, but very few documented extinctions of the plants that rely on them for dispersal. So, this is interesting but may require more discussion.

RESPONSE: We thank the reviewer for this interesting thought. We do not think that our findings contradict documented animal extinctions on islands as we show that animal coextinctions are more likely to occur than plant coextinctions. We added two references to studies of mutualistic networks on islands that report a lack of animal pollinators and seed dispersers in island ecosystems, while network structure is generally maintained despite the lack of animal species (lines 198-200).

The data and methods seem carefully thought out, but I was surprised to see an infrequently-used method of estimating species climatic niches, the OMI. I am unfamiliar with this method and I am not sure how it compares to other methods. It may be a perfectly rigorous and appropriate method, but I would like to see more explanation and justification.

Other methods:

Blonder B, Lamanna C, Violle C, Enquist BJ. 2014. The n-dimensional hypervolume. *Global Ecology and Biogeography* 23:595-609.

Broennimann, O., Fitzpatrick, M. C., Pearman, P. B., Petitpierre, B., Pellissier, L., Yoccoz, N. G., Thuiller, W., Fortin, M.-J., Randin, C., Zimmermann, N. E., Graham, C. H. & Guisan, A. (2012) Measuring ecological niche overlap from occurrence and spatial environmental data. *Global Ecology and Biogeography*, 21, 481-497

Petitpierre, B., Kueffer, C., Broennimann, O., Randin, C., Daehler, C. & Guisan, A. (2012) Climatic niche shifts are rare among terrestrial plant invaders. *Science*, 335, 1344-8.

RESPONSE: As suggested, we added more information on the OMI method (lines 291-299). In addition, we have computed the climatic hypervolume for each plant and animal species using the method proposed by Blonder et al. (2014). The results from both approaches (OMI, hypervolume) are now shown in the manuscript and the results are qualitatively identical (please see revised Fig. 1).

Species distribution models:

What is the sample size for each model? Were absences or pseudo-absences used? How were they selected? How was bias in the sample addressed in the modeling? This can have a big effect on

SDMs. AUC for some groups actually seems a bit low (<0.9) although the literature would classify this as 'good.' Also AUC based on presence versus pseudo-absence has been criticized - were other metrics considered (TSS)?

RESPONSE: The sample size for each species is reported in Supplementary Table 4 (i.e. the number of grid cells with observed presences). We omitted species from the analyses for which no or deficient occurrence data were digitally available (lines 280-281). The species with the lowest number of occurrences (n = 19 grid cells) was Heracleum austriacum (Apiaceae), a range-restricted plant from the Eastern Alps. The median number of occurrences across all species was 815 occupied grid cells per species (lines 281-283). To mitigate sampling bias (i.e. an overestimation of species' absences), we analyzed distribution data on a 50 x 50 km grid although the data are available on a higher spatial resolution (lines 275-276). We further excluded poorly sampled areas from eastern Europe (see Supplementary Figure 1 for the chosen grid and lines 273-275). This ensures a high reliability of the recorded absences for each species. The boosted regression tree analysis uses presence-absence data (information added in lines 308-311). As suggested, we quantified TSS values in addition to AUC values to assess model accuracy for each species (reported in Supplementary Table 4 and explained in lines 311-319). To account for model uncertainty in further analyses, we now weight the mixed-effects models relating biotic specialization and vulnerability to climate change by the respective TSSmax values (see Fig. 1, Supplementary Tables 2-3 and lines 397-401). Results are identical in the analyses accounting for model uncertainty, i.e. biotic specialization and vulnerability of animals were significantly related also in the weighted models.

Although the range size and niche breadth are calculated from atlas data, the change in climate suitability for grid cells is calculated from the modeled current climatic suitability and projected future climatic suitability. So, any discrepancy between current distribution and current modeled distribution is going to add noise to the analysis of vulnerability. Does this require a high degree of accuracy of the SDMs? Can you comment on how this uncertainty affects the conclusions?

RESPONSE: The SDMs are not affected by differences between the recorded and modeled current distribution as we quantified changes in climatic suitability only for each grid cell covered by the species' current European range (defined by the recorded occurrences). We repeatedly clarify this in the revised manuscript (e.g. lines 85-88, lines 325-331).

The SDM-based projections of species' vulnerability to climate change (change in climatic suitability) are really estimates of species exposure to climate change. As pointed out by Schwartz, vulnerability depends on characteristics of the species (niche breadth and biotic interaction among them), and this approach tends to overestimate species' vulnerability to extirpation.

Schwartz MW. 2012. Using niche models with climate projections to inform conservation management decisions. *Biological Conservation* 155:149-156.

RESPONSE: As suggested, we added a few sentences to the manuscript explaining the strength and weakness of this approach according to Schwartz (2012) [see lines 332-338]. We quantified species' exposure to projected changes in climatic conditions to avoid model uncertainties due to species' dispersal ability, non-analogue climates or novel biotic interactions. Nevertheless, we acknowledge the limitations of this approach also in the main text of the manuscript (lines 156-157). Please also note that our approach compares differences in vulnerability to projected climate change among species and the simulation models of species coextinctions are insensitive to absolute vulnerability values because they are based on species' ranks of vulnerability (see Fig. 2 and lines 407-413).

Can you discuss the effect of the coarse scale (50x50 km) of the analysis? Species that occur together within a 50x50km grid cell may not actually interact. Furthermore, projected changes in climatically suitable habitat are sensitive to data resolution (Franklin et al. 2013). So the co-occurrences in the

analysis may be an artifact of scale. How might this affect your conclusions?

FRANKLIN, J., DAVIS, F. W., IKAGAMI, M., SYPHARD, A. D., FLINT, A., FLINT, L. & HANNAH, L. 2013. Modeling plant species distributions under future climates: how fine-scale do climate models need to be? *Global Change Biology* 19(2):473-483.

RESPONSE: The interaction data are based on direct observations between plants and their animal pollinators and seed dispersers recorded in 13 mutualistic networks from central Europe (see Supplementary Table 1). Thus, we derive independent data of biotic specialization (from the empirical interaction networks at a regional scale) and of climatic niche breadth and vulnerability (from occurrence data and projected changes in climatic suitability in the current European species' range). We clarified this in the revised manuscript (lines 88-94, lines 348-350). We deliberately chose a grid size of 50 x 50 km to increase the reliability of absence records for the species distribution models (see above and lines 275-276). As we aimed at quantifying climatic suitability changes in the current species' range, we are confident that this spatial scale is appropriate for our analysis.

What is d' in Table S4?

RESPONSE: The metric d' is a measure of complementary biotic specialization derived from the respective interaction networks (see lines 383-388). The information has now also been added to the respective Table caption.

Reviewer #2 (Remarks to the Author):

In the manuscript "Ecological networks are more sensitive to plant than animal extinction under climate change" Schleuning et al. present a novel way (at least to me) of integrating climate change and species interactions. In particular, they show that animal species with a more specialized interaction nature tend to have smaller geographical ranges, tend to have smaller climatic niches and tend to be the ones most severely affected by climate changes. A similar pattern was not seen for plant species. Furthermore, they look into how primary removal of animals and plants may result in secondary extinctions and conclude that plant removal is more severe than animal removal. To me the former finding (i.e. the relationship between biotic specialization and geographical ranges, climatic niches and climatic vulnerability) seem more interesting because the later could (as far as I can tell, but I might be mistaking) be strongly influenced by the fact that plants tend to have more interaction partners than animals (see also a more detailed comment below). I find the manuscript of a high quality, well written, and very interesting and I think it could be of broad interest for the ecological community. The statistical analyses seem valid and the conclusions seem robust, with appropriate credit to previous work. That said, I think it could benefit from more precise and detailed descriptions at several occasions. See specific comments below.

I'm missing a clear explanation of how the authors selected which interaction networks to include in their dataset. Which criteria were used? Without such justification it may seem as a random sample (maybe based on accessibility) with a strong preference for networks located in Germany.

RESPONSE: We included these networks in the analyses because they report comprehensive data on interaction frequencies between species pairs, consistently recorded by direct plant observations at multiple localities within 13 regions in Central Europe over several months (see addition in lines 348-356). Only a few other quantitative pollination and seed-dispersal networks, recorded with a similar sampling design and intensity, are available globally. The focus on Central Europe is also due to the availability of accurate distribution data for the studied plant and animal taxa, especially for the insect pollinators.

Regarding the interaction network it is not completely clear to me how many species from the original networks that were omitted. In e.g. Weiner et al. (2014) (used three times in the dataset) interaction network consisted of Diptera, Hymenoptera, Lepidoptera and Coleoptera. However, in the current analysis the authors have restricted their focus to Syrphids (one family within Diptera), Hymenoptera and Lepidoptera.

The authors provide some numbers in table S1 that states how many animals that were removed, but do these numbers include all removed species or only removed species belonging to Syrphids, Hymenoptera and Lepidoptera? Some more precision is needed here.

RESPONSE: We have clarified these inaccuracies in our data description. We could not include other than the studied animal taxa (bees, butterflies, hoverflies and birds) because reliable distribution data are only available for these animal taxa. Thus, we excluded other taxa (e.g. other Diptera, Coleoptera) from a few of the original networks (see lines 366-375). This is justified as virtually all empirical interaction networks record interactions only for subsets of the interacting taxa (e.g. focusing only on bee pollinators or avian seed dispersers). The numbers reported in Supplementary Table 1 refer to species with insufficient distribution data from our study taxa (plants; bees, butterflies, hoverflies and birds). This is now clarified in the respective Table caption.

In a similar vein, in line 237-238 the authors write: 'Including species with deficient occurrence data in the calculation of network metrics resulted in qualitatively identical results'. Therefore, when the authors write 'species with insufficient occurrence data' are they then referring to all animals originally included in the network or only Syrphids, Hymenoptera and Lepidoptera with insufficient occurrence data? I think it is important to state if these calculations were done on a subset of the original data or not? Therefore, some more details are needed here.

RESPONSE: This applies to both subsets: all available taxa (e.g. including Coleoptera and other Diptera) and only the animal taxa studied here (see previous response). Thus, sub-setting the networks did not qualitatively change the estimates of biotic specialization for the here studied species. This has also been clarified in the manuscript (see lines 371-375).

Regarding the discussion of why the communities are more sensitive to animal removal than plant removal I think that the authors second mechanism (line 119-124) is probably the primary cause. That is, the reason why communities are more sensitive to plant extinction compared to animal extinction may lie in the differences in linkage level (a difference that is partly documented in line 252-253, and which also can be seen from figure 1). Assuming that most plant species likely have more interaction partners than animal species (and on average receive more total number of visits), and that more animals than plants rely on a single (or a few) plant species (at least that seems plausible from the networks in figure 2), it can be expected that plant removal will inevitable result in more animals experiencing secondary extinctions, than the removal of animals will result in plant secondary extinctions (which I think is also what figure 3 demonstrates when the flexibility is set to 100%). I think this needs to be highlighted and discussed more explicitly around line 119-124.

RESPONSE: We agree with the referee that differences in biotic specialization between plants and animals are related to our finding that plant extinctions have more severe impacts on mutualistic networks than animal extinctions. However, we show that this effect is not due to differences in mean specialization between plants and animals as our random extinction sequences account for such differences between plants and animals (see lines 161-165). That is, the random extinction scenarios are based on the same networks and, thus, the same number of plant and animal species and their respective linkage levels. Since differences between climate change and random extinction are significant in almost all scenarios (see revised Fig. 3), the differences in mean plant and animal specialization are not sufficient to explain the different effects of plant and animal extinction on the

networks. We therefore propose that the different relationships between biotic specialization and vulnerability to projected climate change between plants and animals are the main explanation for this difference (see lines 182-190). That is, animals that are projected to go extinct first have a weaker impact on network structure than the most vulnerable plant species. In the revision, we have clarified this point and now also discuss that the difference between plant and animal extinction is weaker in scenarios where species are less sensitive to coextinction and have a higher flexibility to respond to partner loss (see revised Fig. 3 and lines 169-176, 209-212).

On a more general note I have a hard time understanding why the 'climate change' and the 'random' removal scenarios become more equal when species flexibility increases. Indeed this difference is not at all discussed in the paper and should be included somewhere. If I understand it correctly the 100% flexibility is equal to a situation where species only goes extinct when they lose all their interaction partners, and as such this overrules the authors initial criteria that species goes extinct once they lose 50% of their interactions. I'm not arguing against this 50% reduction methodology, but the results and in particular the contrasting results of having 0% vs. 100% flexibility, and why it becomes closer to random removals, should be discussed more deeply.

RESPONSE: We thank the reviewer for this important comment. As also suggested by reviewer 3, we added more different extinction scenarios to our simulation model and, based on these additional scenarios (see revised Fig. 3), discuss in more detail under which conditions plant extinctions have a more severe effect on network stability than animal extinctions (lines 169-176, 209-212). We also added more information to explain the meaning of the scenario of 100% flexibility (lines 430-434).

Line 6: Missing comma after species?

RESPONSE: Added.

Line 16: What is the difference between 'tolerate' and 'adapt'. Is the former implying no response - in the sense that species are unaffected by the alterations? If so, the sentence "Climate change forces species...to tolerate..." sounds a bit wrong. Moreover, I think this first sentence needs a bit more work to increase readability.

RESPONSE: The sentence was simplified as suggested (line 64).

Line 31-33: Would it make more sense to split it into three distinct test/hypothesis (i) Small realized ranges, (ii) narrow climatic niches and (iii) projected loss in climatic suitability. These tests are also treated separately in the results.

RESPONSE: We do not think that the effects of realized range size and climatic niche breadth on biotic specialization are statistically independent (see their correlations in lines 301-302). We therefore jointly discussed effects of range size and climatic niche breadth on biotic specialization in the previous version. To avoid this ambiguity, we have now removed the analysis of realized range size from the paper as this measure is also more prone to sampling bias than the estimates of realized climatic niche breadth, now derived via the hypervolume and OMI methods (see above in the response to referee 1).

Line 36: I think it would be more informative if the authors specified how many plants, insect pollinators and avian seed dispersers, respectively, they have included (as specified in the Materials and Methods), and not how many they have in total across all three categories. It would also be more precise to state these numbers in the abstract (line 5).

RESPONSE: We added the detailed information as requested (lines 83-85). We do not think that such detail is also needed in the Abstract given the word limit.

Line 41: The authors needs to specify what exactly 'interaction events' covers. I would suggest using 'visitation events' as it could otherwise be incorrectly confused with number of unique interactions. Alternatively, the authors could more precisely specify what they mean.

RESPONSE: We have clarified this statement as suggested (lines 91-94).

Line 49: If I understand the Materials and Methods correctly it really isn't the number of animal partners that the authors are looking at. Maybe it would be more accurate to write what it actually is, viz. 'diversity in partners' (measured with Shannon-Weaver,). Or, at least call it effective partners as done in line 47. Actually, the authors should be more coherent throughout and for example use 'effective numbers of partners' when referring to this metric.

RESPONSE: We have carefully revised the wording throughout the manuscript and now refer to the 'effective number of partners' or the 'diversity of partners' where this is more appropriate.

Line 49-51: I guess a similar statement could be made for 'climatic niche breadth'?

RESPONSE: We have excluded the analysis of realized range size and now only refer to climatic niche breadth here and elsewhere.

Line 72: Should it be 'effective partners'.

RESPONSE: Changed as suggested.

Line 119: I suggest that the authors start the sentence with 'Second', so that it follows the logic of presenting the two different mechanisms.

RESPONSE: We added 'second mechanism' to the sentence in line 185.

Line 125-126: I'm not sure I understand this sentence. Which ecological function are the authors refereeing to, and which results are supporting this?

RESPONSE: We refer to pollination and seed dispersal by animals (now specified, see line 192) as these functions are mediated by the studied plant-animal interaction networks.

Line 143: Delete 'a'?

RESPONSE: Changed as suggested.

Line 168: Maybe the authors should add that all networks were quantitative (although this becomes apparent further down in the Materials and Methods). The authors might also add this information earlier in the manuscript.

RESPONSE: Changed as suggested.

Line 248-250: Based on the description I'm a bit unsure whether the metric 'effective number of partners' is defined exactly as e^H , or if it is some other measure based on e^H which can be found in Bersier et al. 2002?

RESPONSE: Yes, the metric corresponds to e^H according to Bersier et al. 2002.

Line 252-253: Strictly speaking, when using e^H it is not possible to say that plants interacted with more partners than animals, as compared to just interacting more equally with the partners they actually have. I suggest rephrasing to something like: '...plants had a higher diversity of partners....'

RESPONSE: We carefully checked and rephrased here and throughout the manuscript (see also our previous response to a similar comment above).

Line 254: Should 'a' be removed?

RESPONSE: Removed as suggested.

Line 259-265: Did the authors allow both random intercept and slope in their linear mixed model as suggested by for example Schielzeth & Forstmeier (2009). Please explain.

Line 266-269: I'm not completely sure what the authors have tested here, and how it deviates from the linear mixed model procedure described in line 259-265. Please provide some more details.

RESPONSE: We added the requested information to improve clarity (lines 395-397). Models across all networks and including plant and animal species simultaneously only included random intercept effects (see Fig. 1 and Supplementary Table 2). We additionally ran separate models for plant and animal species including additionally random slope effects of the respective predictor variable on network identity (lines 403-406). These models show that the estimated effects of the respective climatic predictor variable on biotic specialization were similar across the individual networks (see Supplementary Table 3).

Figure 2

In the figure legend the authors refer to 'light shaded' and 'dark shaded' as a scale that goes from high to low vulnerability. However, in the figure it is the interaction that are coloured end not the species, which is a bit confusing. Although this colouring of the interactions could be meaningful I would suggest arranging the species from low to high vulnerability. That would make the removal sequence much more apparent at first sight, and would likely make it more convenient to relate the colour of the interactions with the scale-bar at the bottom.

Furthermore, I would suggest renaming the labels of the sensitivity plots so that on the x-axis it would be 'plants removed' or 'animals removed', and on the y-axis it would be 'animals persisting' or 'plants persisting'. That would greatly aid readers not familiar with such plots.

Finally, the authors could even consider labelling the two rows of the bipartite plots with 'animals' and 'plants'.

RESPONSE: We have reordered the species according to species' vulnerability as suggested. We agree that it is now easier to match the scale bar at the bottom with the species order in the network. We also relabeled the other plots as suggested.

References:

Bersier, L.-F., Banašek-Richter, C. & Cattin, M.-F. (2002) Quantitative descriptors of foodweb matrices. *Ecology*, 83, 2394-2407.

Schielzeth, H. & Forstmeier, W. (2009) Conclusions beyond support: overconfident estimates in mixed models. *Behavioral Ecology*, 20, 416-420.

Weiner, C.N., Werner, M., Linsenmair, K.E. & Blüthgen, N. (2014) Land-use impacts on plant-pollinator networks: interaction strength and specialization predict pollinator declines. *Ecology*, 95, 466-474.

Reviewer #3 (Remarks to the Author):

Effects on climate change on individual species have been documented in many cases. More recently, there has been a growing interest in the effect of climate changes on interactions among species. Iconic examples are phenological mismatches between herbivores and plants, or predators and prey. However, most of these studies focused on a few case examples, and there are only few studies that looked at effects of climate change (or other environmental changes) on large interaction networks (e.g. Peasre et al 2013).

In their manuscript, Schleuning et al. addressed how interaction networks (using empirical data on plant-pollinator and plant-seed-dispersal networks) could be affected by climate change. They find in a combination of species-distribution models and ecological network analysis that different groups of species and their interactions are differently vulnerable to be affected by climate change. The main finding is that animal species that are specialized on few plants are more vulnerable to climate change, while biotic specialization of plants is not related to vulnerability in the context of climate change. Consequently, animal species are supposed to be more likely affected by co-extinctions than plant species.

This finding is interesting, and has to my knowledge not been shown in such a comparison of several interaction networks. However, the finding is by itself not very surprising, see for example classic work by Warren et al. 2001. The models used in the study make the relatively strong assumption that the interaction-link strengths are all equal, and-even stronger assumption-that links between different functional groups are directly comparable. I think this is clearly not the case, and should at least be stated. The finding that removing links has stronger effects on animals depending on plants than vice-versa depends on the assumption that the significance of the links is a priori the same (key finding line 108-109). I am not sure if that can indeed be assumed; at least it is a very strong assumption and this limitation should be better discussed.

RESPONSE: We thank the reviewer for this thoughtful comment. We agree that our analyses, as basically all analyses of empirical ecological networks, assume that interaction frequencies adequately describe the reciprocal dependencies between species (e.g. see lines 158-159 where we state this assumption). Our models account for potential changes in interaction frequencies (quantitative effects), but do not account for potentially variable qualities of interactions between different groups of species (e.g. see lines 192-196). However, interaction frequencies between species have been shown to be indeed related to impacts of animals on plants and vice versa (see lines 358-361 and the study by Vázquez et al. 2012 cited there). In the revised manuscript, we acknowledge that our findings are contingent on this assumption (lines 222-223) and propose experimental studies for testing the findings of our simulation model (lines 234-236).

We do not agree that our main finding has been shown before in other studies. For instance, the study by Warren et al. 2001 does not account for biotic specialization of butterfly species, but does highlight the importance of habitat specialization. Here, we investigate for the first time species' coextinction risks under climate change for plants and animals directly linked via interactions in plant-animal mutualistic networks.

While the text reads generally well, I found it a bit narrative and I was hoping to see more of the underlying support/link to the data to better support the conclusions. I do not say this support is not there, just that it should be much more specifically shown. For example, data points in the figures (e.g. Fig. 1) should be added, not only model fits. Also, much more details of the 13 interaction networks could be given. It would be helpful to have a visual depiction of all individual networks, even in the main article, and a summary table of the key network parameters generally used in mutualistic networks (e.g., nestedness, modularity, number of links, etc.). For example, the findings

reported for one interaction matrix (Fig. 2) should be given for all 11 remaining ones as well (supplement); also, it is nowadays standard that the interaction matrices are provided in a machine-readable form. The interaction strengths but also limitations of the findings with respect to the individual interaction matrices should be discussed.

RESPONSE: We carefully followed these suggestions. We now show partial residual plots with the individual data points ($n = 295$ plant and 414 animal species) in the revised Fig. 1. We also provide secondary extinction plots for all 12 remaining networks in an additional Supplementary Figure 3. Furthermore, we added more information to Supplementary Table 1 now also reporting some key network properties (i.e. number of links, connectance, nestedness, modularity, vulnerability, generality, H_2). We do not think that network plots of all 13 networks would be informative. This addition would also require a lot of space, e.g. the largest network comprises almost 300 species and five networks comprise more than 100 animal species. Nevertheless, we could of course provide such a multi-panel plot for the Supplement if deemed necessary. We would also like to stress that all main analyses have been conducted and reported across the 13 individual networks (e.g. see Fig. 1 and Fig. 3). Thus, the focus of this analysis is not on the individual networks, but on the consistent patterns across networks (see Supplementary Table 3). Similarly, we do report all data at species level (e.g. biotic specialization, climatic niche breadth, vulnerability to projected climate change) in Supplementary Table 4. The individual networks are published in the original studies (see the sources provided in Supplementary Table 1) and would be available upon request from the respective data holders.

Specific additional comments:

- There is already quite a bit of literature that has been analyzing co-extinctions in plant-insect interaction networks, looking for example at how range size etc. are affecting the extinction cascades (e.g., Peasre et al. 2013 Ecology; this and other work might be worth to be cited as it is using similar approaches). Also, one of the other main findings, namely that specialized animal species may be more vulnerable to climate change than generalists, is not novel (e.g., Warren et al. 2001 Nature). These findings should be better discussed in the context of previous work.

RESPONSE: We thank the reviewer for these suggestions. We now cite the interesting study by Pearse and Altermatt (2013) that shows the importance of butterfly coextinctions in response to host plant extinctions. However, this study did not test the importance of coextinctions under climate change and did not analyze quantitative (frequency-weighted) ecological networks, but is based on a literature survey of host plants of butterfly species. While this is a valuable approach, it is strikingly different to our study that accounts for differences in interaction frequencies both in the estimation of biotic specialization and in the simulation models of species coextinctions. Nevertheless, as suggested in this paper, we briefly discuss the importance of direct vs. indirect effects of climate change (or other drivers) on species extinctions (lines 228-231). We now also cite a study by Pellissier et al. 2013 (line 78) that combined species distribution models with a binary food web between butterflies and their host plants (derived from a literature survey and not from direct, frequency-weighted observations as in our study). We do not agree that the study by Warren et al. 2001 is closely related to our work as it does not actually test for biotic specialization of butterfly species, but is based on an a priori binary classification of habitat specialization of butterfly species. We already cite and refer to the studies by Schweiger et al. 2012 and Stewart et al. 2015 that show that species ranges and range shifts are related to biotic specialization for antagonistic plant-insect interactions. In extension to these earlier studies, we here present the first study for mutualistic plant-animal interactions including plants and their bee, butterfly and hoverfly pollinators and avian seed dispersers (lines 116-118). We further directly link quantitative interaction networks to species' current climatic niche breadth and vulnerability to projected climate change. We combine this information in the first simulation model of plant and animal coextinctions under projected climate change. We stress the

novelty of our study at several places in the manuscript (e.g. lines 74-78, 116-118, 142-146, 236-238).

- It was not clear to me if the observed correlation between realized range size and effective number of interaction partners referred to effective number of local interaction partners, or across the range (lines 46 to 48). It would be possible to have few interaction partners locally, but many regionally (i.e., different partners at different localities) or to have many partners locally, but everywhere the same. The consequences with respect to extinction cascades are very different: in the former case a few losses are dramatic, while in the latter the networks are highly buffered. Please clarify.

RESPONSE: We always refer to species' biotic specialization within each regional mutualistic network. We have clarified this in the revised manuscript (e.g. lines 91-94, 98-100, 124-126). The idea to quantify turn-over of interaction partners across regions and to use such information for informing simulation models of network rewiring and extinction cascades is interesting, but would be beyond the scope of this study.

- The study looked at several (13) interaction networks, which is laudable. However, they are often treated as one ("720 plant and animal species in pollination and seed-dispersal networks"). I think it would be better to refer more to the actual number of networks, and their structure, than on the total number of species only (e.g., abstract). Especially in the main text these networks are a bit treated as a "black box", and the discussion of the results is often at a meta-level, where details to the individual networks is lacking. Also, I missed more detailed information and description of these networks. It would be relatively easy to visualize these 13 networks. Also, the limitations of these interaction networks should be clearly discussed: do I understand it correctly that interactions were inferred from co-occurrences in 50x50 plots? This is a rather strong assumption for organisms with an often much smaller mobility.

RESPONSE: We deliberately refer to the patterns across networks and do not aim at analyzing peculiarities of the individual networks that have been described elsewhere (see references given in Supplementary Table 1). Please note that the individual networks showed largely consistent patterns in the analyses of biotic specialization against climatic niche breadth and vulnerability to projected climate change (Supplementary Table 3) and also in the simulations of species coextinctions (see new Supplementary Fig. 3). Please also note that the species (and not the networks) are the replicates in our analysis as all relevant measures (i.e. biotic specialization, climatic niche breadth, projected vulnerability) are derived for each individual species, while we account for network identity in all analyses (e.g. see revised Fig. 1). Analyses of the overall network structure are not within the scope of this study and we do not think that adding 13 network plots would provide added value to the manuscript (see also our response to a similar comment above). All interaction networks have been recorded by direct plant observations at multiple localities within 13 regions over several months. Thus, interaction frequencies describe observed frequencies of interactions between plant and animal species and are not based on species co-occurrences. We have clarified this in the manuscript (lines 88-94, 348-356).

- The study makes the strong assumption that a loss of interaction in a pollinator-plant network causes subsequent extinctions. Is this realistic? I think this may be realistic for some bee species which are relatively specialized pollinators, but much less so for butterflies. This aspects needs much better justification. Butterfly pollinators are notoriously unspecific in their choice of nectaring plants, and to my knowledge there is relatively few plant species in Central Europe that are depending on a few key pollinators, and very few or even none of them is a "butterfly plant". Thus, it is empirically not very strongly supported assumption that loss of pollination interactions results in a loss of species. This at least needs to be discussed, or different levels of interaction strengths could be introduced in the model.

RESPONSE: The notion described by the reviewer is quantitatively tested in our analysis. Plant species and animal species exhibit different degrees of biotic specialization in mutualistic networks. We test whether these differences in biotic specialization are related to a species' climatic niche breadth and projected vulnerability to climate change. We are aware that our simulations are based on the assumption that interaction frequencies are related to the reciprocal functional dependencies between species (see our detailed response above). Moreover, our findings could be sensitive to different scenarios of species' rewiring and sensitivity to coextinction. As suggested, we therefore added more simulation scenarios accounting for differences in rewiring (constrained vs. unconstrained rewiring, see below) and coextinction thresholds (after 25, 50 and 75% of interaction loss). We found similar results for all scenarios (see revised Fig. 3 and Supplementary Fig. 4) and now discuss the specific differences between scenarios in the manuscript (e.g. see lines 169-176, 207-212).

- Some of the studied species (especially butterflies) are not only having mutualistic interactions with plants, but also antagonistic interactions in their larval stage. While nectaring/pollinating is an important interaction, it is likely much less essential for a butterfly's persistence than the antagonistic interaction in the larval stage, and is arguable also much less specific than for example interactions of bees, which are known to me much more host-specific pollinators, while butterflies are more generalist pollinators.

RESPONSE: We agree with the reviewer that antagonistic interactions in the larval stage can be very important, especially for some butterfly species. However, such effects have been studied previously (as also stressed by the reviewer earlier), while we focus here on the not yet studied impacts of climate change on mutualistic interactions between plants and adult animals. Nevertheless, we discuss in the paper that the dependence on plants may be particularly strong for animal species that depend on plants both in the larval and adult stage (see lines 225-228).

- The study assumes that interactions are fixed and can't be "rewired" (lines 100-103). While I can see that this assumption makes simplifies the data analysis, I am not convinced that it actually reflects reality: in the context of climate change novel interaction partners show up, and plant-insect networks have lately been shown to be rather flexible in incorporating novel trophic interactions. Given that the study is based on models and extinction scenarios, it would be relatively easy to incorporate also possible host switches, and do so under different assumptions. I would strongly recommend to do additional work showing how robust/dependent the results are on assumptions regarding host-use flexibility.

RESPONSE: We thank the reviewer for this interesting suggestion. As suggested, we now include an additional scenario of unconstrained rewiring to new partners. Thus, we simulate the rewiring of interactions (i) to all persisting partners (constrained rewiring) and (ii) to all species persisting in the network (unconstrained rewiring). Both approaches result in very similar results and stress the robustness of our main finding that animal coextinctions are more likely to occur than plant coextinctions under climate change (see revised Fig. 3). We do not account for potentially new species entering the networks as this would require additional assumptions, e.g. on the dispersal ability of species. We clearly state this assumption in the manuscript (lines 156-157, 222-223).

Literature mentioned:

Pearse et al. 2013 Extinction cascades partially estimate herbivore losses in a complete Lepidoptera-plant food web. *Ecology* 94, 1785-1794.

Warren et al. 2001 Rapid responses of British butterflies to opposing forces of climate and habitat change. *Nature* 414, 65-69.

Reviewers' Comments:

Reviewer #1 (Remarks to the Author):

The revisions to the manuscript, including new analyses, and the response to reviewers comments, have thoroughly addressed my previous comments. I think the authors have done a fine job.

Reviewer #2 (Remarks to the Author):

I find the revised manuscript entitled “Ecological networks are more sensitive to plant than animal extinction under climate change “greatly improved. Schleuning and co-authors have addressed all the major points I raised during the last round of revision.

Among the improvements (in response to not only my comments, but also the ones made by the other reviewers) are

- 1) More precise and detailed explanations of the methods and procedures applied in the manuscript, making the whole manuscript more logic to follow.
- 2) The authors have added another level of rewiring (allocating lost interactions to persisting species) besides the one already included (allocating lost interactions to persisting partners)
- 3) Both figure 1 (where the authors have added the actual data points) and figure 2 (where the authors have rearranged nodes, and added more intuitive labels) have, in my opinion, been greatly improved.
- 4) Also the revised figure 3 has added more depth and detail to the paper.

Overall, I still find the manuscript of a high quality, well written, and very interesting. I only have a few minor comments that the authors might find useful

Abstract

Line 54: I still don't think the authors can say much about the number of interaction partners as compared to saying something about the diversity in partners. Maybe a bit of rewording to remove this 'interact with few plant species'. I suppose the authors mean 'effective number of partners' or something similar.

Main text

Line 83-85: I like the added details

Line 99: Remove 'their'?

Line 106: Maybe a bit of rewording to make it more precise. As it is now, it sounds like the hypothesis was confirmed, and then as a side note it is mentioned that it did not match for plants species. Maybe simply start the sentence with: ‘For animals the hypothesis....’ or some other solution.

Line 236-238: I think the statement in line 236-238 is particular strong and well-suited for the findings in this manuscript. I will leave it to the authors to decide, but I think this sentence, or a modified version, would be very appropriate for the abstract.

Line 353: The authors write that observations were recorded at multiple localities within each region. I’m wondering what lies in the word “multiple”. Maybe it would be worth adding this detail to table S1 - that is, how many localities were surveyed within each region. That would also give the reader an impression of the extent of the survey in each of the included studies.

Line 441-443: It is not completely clear from the description whether allocations of lost interactions were allowed during the random removal scenario. I assume such rewiring was allowed, as it otherwise would not make sense to compare the two. However, please add some more details.

Line 449: It might be confusing for readers (especially readers unfamiliar with the network jargon) that the authors use “vulnerability” for two very different measures. In table S1 you have “vulnerability” as the mean number of effective animal partners per plant species, while in other connections (e.g. line 56, 76, 326, 449 etc.), the authors use “vulnerability” to describe the sensitivity of species to the projected climate changes.

Figure 1

I would suggest adding e^H as part of the y-label so that fast readers immediately know which measure you refer to, or alternatively, at least mention it in the legend after ‘biotic specialization’.

Reviewer #3 (Remarks to the Author):

With great pleasure I read the revised version of the manuscript, and I would like to congratulate the authors for their careful integration of the comments. I mostly looked at how my comments (reviewer 3 in first round) were addressed, and I am very happy with it.

Specifically, I very much appreciate the data points added to figure 1, I think it profited a lot, and looks now much clearer and gives a better feeling for the data. Also, the additional

supplementary Fig. 3 is very nice, and I am very happy that the authors could do that based on my suggestions. I recognize that they did so in a careful way, which I appreciate. I personally would still visualize the interaction networks, but this may really be a matter of taste, so at that point I respect the authors decision and wouldn't insist on it.

Secondly, I really appreciate that the authors picked up my comment on the rewiring of interactions, and to see consequences of this. They did this in a thorough manner, and I think the findings are good and robust. Thus, while my earlier comments definitely created some additional work for the authors, I think the manuscript profited from how they addressed it.

I am happy with all other responses and how they were addressed and I am looking forward to see this study published.

Point-by-point response to reviewers

We thank the reviewers for their very positive reception of our revised manuscript. Below we provide an itemized response to each of your comments during the second round of review; our responses to each comment are written in italics.

Reviewer #1 (Remarks to the Author):

The revisions to the manuscript, including new analyses, and the response to reviewers comments, have thoroughly addressed my previous comments. I think the authors have done a fine job.

Response: We thank the reviewer for this positive feedback.

Reviewer #2 (Remarks to the Author):

I find the revised manuscript entitled 'Ecological networks are more sensitive to plant than animal extinction under climate change' greatly improved. Schleuning and co-authors have addressed all the major points I raised during the last round of revision. Among the improvements (in response to not only my comments, but also the ones made by the other reviewers) are

1) More precise and detailed explanations of the methods and procedures applied in the manuscript, making the whole manuscript more logic to follow.

2) The authors have added another level of rewiring (allocating lost interactions to persisting species) besides the one already included (allocating lost interactions to persisting partners)

3) Both figure 1 (where the authors have added the actual data points) and figure 2 (where the authors have rearranged nodes, and added more intuitive labels) have, in my opinion, been greatly improved.

4) Also the revised figure 3 has added more depth and detail to the paper.

Overall, I still find the manuscript of a high quality, well written, and very interesting. I only have a few minor comments that the authors might find useful

Response: Thank you for these supportive comments on the revised manuscript.

Abstract

Line 54: I still don't think the authors can say much about the number of interaction partners as compared to saying something about the diversity in partners. Maybe a bit of rewording to remove this 'interact with few plant species'. I suppose the authors mean 'effective number of partners'; or something similar.

Response: We replaced 'few plant species' by 'a low diversity of plant species'.

Main text

Line 83-85: I like the added details

Response: Thank you.

Line 99: Remove 'their'?

Response: Changed as suggested.

Line 106: Maybe a bit of rewording to make it more precise. As it is now, it sounds like the hypothesis was confirmed, and then as a side note it is mentioned that it did not match for plants species. Maybe simply start the sentence with: 'For animals the hypothesis' or some other solution.

Response: Changed as suggested. The sentence has been moved to the beginning of the now separate Discussion section.

Line 236-238: I think the statement in line 236-238 is particular strong and well-suited for the findings in this manuscript. I will leave it to the authors to decide, but I think this sentence, or a modified version, would be very appropriate for the abstract.

Response: We followed this suggestion and have moved a modified version of this sentence to the end of the Introduction where we added a paragraph that highlights the main findings and conclusions. We had to shorten the Abstract due to the word limit.

Line 353: The authors write that observations were recorded at multiple localities within each region. I'm wondering what lies in the word 'multiple'. Maybe it would be worth adding this detail to table S1 - that is, how many localities were surveyed within each region. That would also give the reader an impression of the extent of the survey in each of the included studies.

Response: We could not obtain the respective information for all of the original networks and have removed 'that all networks were recorded at multiple localities' from this sentence. This sentence now only refers to the number of sampling months that is provided for each network in Supplementary Data 1. Nevertheless, we clarified in one of the following sentences that 'most networks were recorded on repeated visits at several localities within each region'.

Line 441-443: It is not completely clear from the description whether allocations of lost interactions were allowed during the random removal scenario. I assume such rewiring was allowed, as it otherwise would not make sense to compare the two. However, please add some more details.

Response: We added the requested information. The same scenarios of coextinction, rewiring and flexibility were modelled and compared between climate change and random extinction.

Line 449: It might be confusing for readers (especially readers unfamiliar with the network jargon) that the authors use 'vulnerability' for two very different measures. In table S1 you have 'vulnerability' as the mean number of effective animal partners per plant species, while in other connections (e.g. line 56, 76, 326, 449 etc.), the authors use 'vulnerability' to describe the sensitivity of species to the projected climate changes.

Response: We replaced 'vulnerability' by the 'mean number of effective animal partners' and 'generality' by the 'mean number of effective plant partners' in Supplementary Data 1.

Figure 1

I would suggest adding e^H as part of the y-label so that fast readers immediately know which measure you refer to, or alternatively, at least mention it in the legend after 'biotic specialization'.

Response: We added the requested information to the figure caption.

Reviewer #3 (Remarks to the Author):

With great pleasure I read the revised version of the manuscript, and I would like to congratulate the authors for their careful integration of the comments. I mostly looked at how my comments (reviewer 3 in first round) were addressed, and I am very happy with it.

Specifically, I very much appreciate the data points added to figure 1, I think it profited a lot, and looks now much clearer and gives a better feeling for the data. Also, the additional supplementary Fig. 3 is very nice, and I am very happy that the authors could do that based on my suggestions. I recognize that they did so in a careful way, which I appreciate. I personally would still visualize the interaction networks, but this may really be a matter of taste, so at that point I respect the authors decision and wouldn't insist on it.

Secondly, I really appreciate that the authors picked up my comment on the rewiring of interactions, and to see consequences of this. They did this in a thorough manner, and I think the findings are good and robust. Thus, while my earlier comments definitely created some additional work for the authors, I think the manuscript profited from how they addressed it.

I am happy with all other responses and how they were addressed and I am looking forward to see this study published.

Response: We thank the reviewer for this very encouraging feedback.